# *Bézier Meets Diffusion*: ROBUST GENERATION ACROSS DOMAINS FOR MEDICAL IMAGE SEGMENTATION

## ABSTRACT

Training robust learning algorithms across different medical imaging modalities is challenging due to the large domain gap. Unsupervised domain adaptation (UDA) mitigates this problem by using annotated images from the source domain and unlabeled images from the target domain to train the deep models. Existing approaches often rely on GAN-based style transfer, but these methods struggle to capture cross-domain mappings in regions with high variability. In this paper, we propose a unified framework, *Bézier Meets Diffusion*, for cross-domain image generation. First, we introduce a Bézier-curve-based style transfer strategy that effectively reduces the domain gap between source and target domains. The transferred source images enable the training of a more robust segmentation model across domains. Thereafter, using pseudo-labels generated by this segmentation model on the target domain, we train a conditional diffusion model (CDM) to synthesize high-quality, labeled target-domain images. To mitigate the impact of noisy pseudo-labels, we further develop an uncertainty-guided score matching method that improves the robustness of CDM training. Extensive experiments on public datasets demonstrate that our approach generates realistic labeled images, significantly augmenting the target domain and improving segmentation performance.

## 1 INTRODUCTION

Deep learning models rely on supervision from training data to achieve superior performance across various fields. However, the limited scope of training datasets often leads to substantial domain gaps between curated training samples and real-world data. These gaps hinder the deployment of deep models in practical applications. Unsupervised domain adaptation (UDA) (Chen et al., 2019; Huo et al., 2018; Jiang et al., 2020; Liu, 2019) has emerged as a promising solution to this challenge, which leverages labeled data from a source domain and transfers knowledge to a target domain that contains only unlabeled data. The key idea is that, with carefully designed adaptation strategies, one can bridge the domain gap and train an effective target-domain model without the costly and time-consuming process of manual annotation.

For the UDA task, one popular direction is GAN-based style translation (Liu et al., 2017; Zhu et al., 2017) (see Fig. 1(a)). These methods either transform target domain images to the source domain style, enabling the source-trained model to achieve better segmentation performance on style-transferred target images, or convert source domain images to the target domain style, allowing the transferred images to augment the training set and enhance target domain segmentation performance. Aside from common issues such as instability in GAN training, these methods often produce suboptimal images, especially for regions with large variations. For example, lesions occur less frequently, occupy smaller areas, and exhibit heterogeneity in location, shape, and texture. They are difficult to segment for domain-translation methods, as it is hard to establish reliable correspondences across lesions from different domains and learn corresponding style mapping (Wang et al., 2024).

To overcome the limitations of existing GAN-based domain translation methods, we introduce a novel Bézier-curve-based style transfer approach. Prior GAN-based methods typically assume a completely free-form transformation (Zhu et al., 2017; Hoffman et al., 2018; Chen et al., 2020; Jiang et al., 2020), which can lead to unstable training and poor generalization. In contrast, our method employs a learnable Bézier-curve-based transformation function (termed *Bézier adaptation*) that introduces nonlinearity into the mapping while constraining the degrees of freedom. This design strikes a balance between flexibility and regularization, thereby improving both learning efficiency

Figure 1: The comparison between (a) GAN-based style transfer methods and (b) our CDM-based augmentation framework.

and generalizability. Furthermore, the transformation is optimized by maximizing feature-space similarity between source and target images, effectively leveraging the strong representational power of deep neural networks. An illustration of this framework is provided in Fig. 2(a).

To further reduce domain gaps, we propose learning a conditional diffusion model (CDM) within the target domain, enabling the generation of labeled images to augment training. These diffusion models generate target domain images conditioned on source domain segmentation masks, based on the classic UDA assumption that the segmentation label distribution is the same across domains. These labeled synthetic images, together with original unlabeled images, are used to train a strong target domain segmentation model. Please see Fig. 1(b) for an illustration. Recall that we do not have ground truth labels in the target domain, and the key challenge is how to train a CDM in the target domain without segmentation labels. A naive solution is to train an initial segmentation model in the source domain, and do inference on target domain images to generate pseudo-labels. However, due to the domain gap, these pseudo-labels are often noisy, thus impairing the training of CDM.

To improve the robustness against the noise in the conditional labels, we further propose a novel uncertainty-guided training of CDM. Our key observation is that, given an imperfect segmentation model, we should not only rely on its segmentation prediction but also on other valuable outputs. In particular, the uncertainty information provides valuable complementary information to the noisy pseudo-labels. Furthermore, we should consider not only the most likely prediction of the segmentation model, but also its secondary prediction, tertiary prediction, etc. In noisy settings, these labels are still likely to match the true label. Based on such observation, we propose a novel uncertainty-guided score matching loss for the CDM training, which incorporates all predictions of the segmentation model holistically. In other words, we train the CDM using the output of the segmentation model, rather than the segmentation alone. See Fig. 2(b).

It is worth noting that our novel training strategy does not change the CDM architecture. At the application stage, i.e., when being used to augment the target domain, the CDM still only requires a single segmentation mask as its conditional input. This robust-to-noise training method can certainly be generalized to many other tasks, beyond UDA. The proposed Bézier-curve-based style transfer method not only enhances UDA performance by reducing domain gaps across different modalities but also improves the quality of uncertainty maps generated by the initial segmentation model trained on the source domain. In summary, our contributions are:

- We propose a Bézier-curve-based style transfer method, guided by feature-space similarity, to reduce domain gaps across different imaging modalities.
- We introduce a novel UDA segmentation framework that leverages a CDM to generate labeled synthetic images, thereby augmenting the target domain and enabling the training of a high-quality segmentation model.
- To ensure robust CDM training in the presence of noisy pseudo-labels, we develop an uncertainty-guided learning strategy that mitigates the impact of label imperfections.
- Our *Bézier Meets Diffusion* framework can be combined with any existing methods to boost their performance, as demonstrated in our experiments.

## 2  RELATED WORK

**Unsupervised domain adaptation** (UDA) for segmentation has been extensively studied as a cost-effective solution to the challenge of acquiring high-quality segmentation annotations (Hoffman et al., 2018; Kim & Byun, 2020; Li et al., 2019; Tsai et al., 2018; Zhao et al., 2024; Zhang et al., 2021), particularly in the context of medical imaging, where manual annotations demand significant

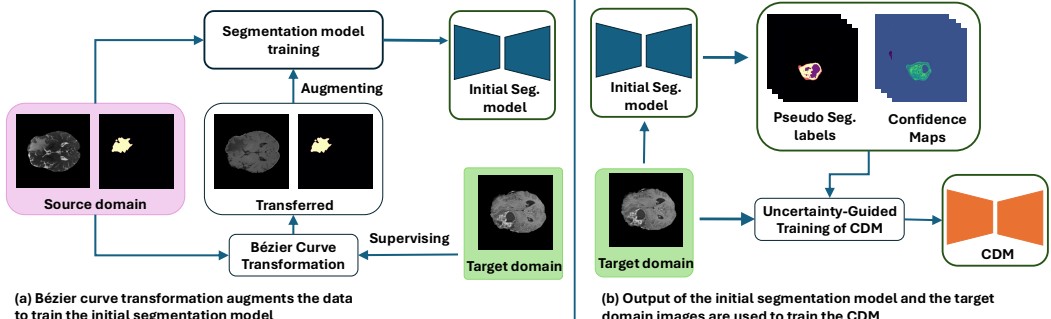

Figure 2: The overview of our contributions: Bézier adaptation and uncertainty guided CDM.

expertise and resources (Chen et al., 2020; Dorent et al., 2020; Huo et al., 2018; Jiang et al., 2020; Liu, 2019; Zhang et al., 2018). Self-training (Brüggemann et al., 2023; Hoyer et al., 2022; Xie et al., 2023; Wu et al., 2021; Shen et al., 2023) is widely utilized in UDA due to the substantial performance gains achieved through pseudo-labeling. However, the gain of self-training is limited due to two aspects: First, the quality of pseudo-labels can be degraded due to the domain gap. Second, the available target domain images determine the performance upper bound of UDA methods. In contrast, our framework leverages the generative capacity of CDM, enabling us to generate an unlimited number of labeled target domain images, irrespective of the availability of target domain samples.

**GAN-based style transfer** (Arjovsky et al., 2017; Goodfellow et al., 2014; Karras et al., 2019; Mao et al., 2017) mitigates domain gaps by adapting the image style of the source domain to match that of the target domain. This enables the segmentation model trained on style-transferred images to achieve improved performance in the target domain (Hoffman et al., 2018; Jiang et al., 2020; Zhu et al., 2017). However, the effectiveness of these domain style transfer methods is limited. Achieving a thorough and precise style transformation is highly challenging. In tumor regions, style transfer can be particularly problematic and may even have a counterproductive effect due to the sparsity and location variability of tumor regions. Although methods like CyCADA (Hoffman et al., 2018) use semantic consistency loss to force the translated image to have the same segmentation map as before, the need to use multiple loss functions impairs the model's ability to depict tumor regions.

**Diffusion models** (Dhariwal & Nichol, 2021; Ho et al., 2020; Nichol & Dhariwal, 2021; Song et al., 2021) are widely applied to vision tasks due to their ability to produce high-quality samples. For the segmentation task, diffusion models are applied to both natural images (Amit et al., 2021; Chen et al., 2023; Ji et al., 2023; Peng et al., 2023) and medical images (Wu et al., 2024b; Wolleb et al., 2022; Li et al., 2024; Xu et al., 2025). In terms of domain adaptation, Peng et al. (2023) proposes a diffusion-based image translation framework guided by pixel-wise semantic labels for semantic segmentation. In Wang & Li (2023), the diffusion model is utilized as an encoder to learn domain-invariant representations, facilitating UDA in medical image segmentation. To the best of our knowledge, we are among the first to leverage conditional diffusion models for directly augmenting the target domain with high-quality labeled synthetic images.

## 3 METHOD

Our *Bézier Meets Diffusion* framework consists of a Bézier-curve-based style transfer module (*Bézier adaptation*, Section 3.1) and an uncertainty-guided conditional diffusion model (CDM) (Section 3.2). In Bézier adaptation, Bézier curves are used to model the mapping between different modalities, with feature-space similarity guiding the evaluation and optimization of mapping quality. By manipulating the control points of the Bézier curves, the mapping can be flexibly adjusted so that source-domain images more closely approximate the target domain. These transferred images serve as effective augmentations, enabling the training of stronger target-domain models.

As shown in Fig. 1(b), our method trains a CDM for the target domain. Leveraging the assumption of shared label distribution across domains, the CDM uses a conditional segmentation mask from the source domain as input and generates a corresponding labeled synthesized image in the target domain. These images are used to augment the source domain data, and they can be used together with the unlabeled target domain images for UDA training to obtain the final segmentation model for the target domain. As an augmentation method, our method can naturally be incorporated with any existing UDA methods. To train the CDM, we use target-domain images paired with pseudo-labels

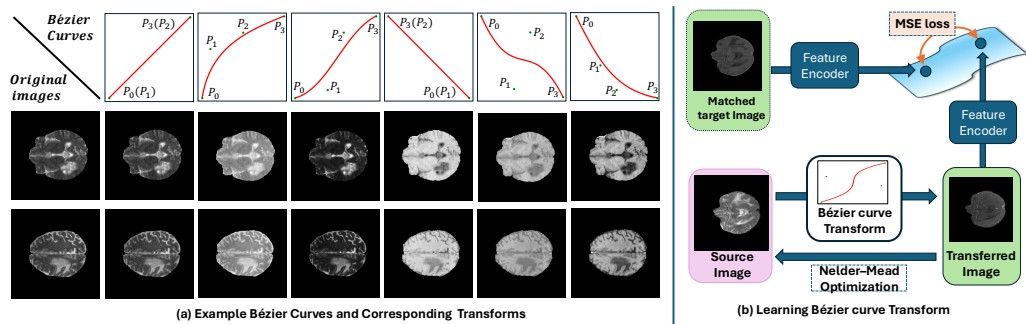

Figure 3: The overview of Bézier-curve-based style transfer. (a) illustrates the impact of varying control points on Bézier curves and their effect on medical image styles. (b) is the control points optimization process of Bézier adaptation.

produced by an initial, imperfect segmentation model (e.g., one trained solely on the source domain). The overview of our framework can refer to Fig. 2.

### 3.1 FEATURE-SIMILARITY OPTIMIZED BEIZER-CURVE-BASED STYLE TRANSFER

**Bézier curves** are particularly suitable for image transformation because they (1) enable a nonlinear mapping between pixel value distributions; (2) are governed by a small number of parameters; and (3) offer sufficient flexibility to model diverse real-world transformations. Specifically, we focus on cubic Bézier curves as a transformation function from source domain intensity range $[0, 1]$ to target domain intensity range $[0, 1]$ (assuming a normalized range). A Bézier curve of degree $n$ is a parametric curve defined by $n + 1$ control points $\{P_i\}_{i=0}^n$, with each point $P_i = (x_i, y_i)$ contributing to the overall curve shape. The curve is expressed using the Bernstein polynomial basis as $B(t) = \sum_{i=1}^n \binom{n}{i}(1 - t)^{n-i} t^i P_i, t \in [0, 1]$ (Farin, 2002). In this work, we consider the cubic case ($n = 3$), where the curve is determined by four control points: two end points $P_0$ and $P_3$ that anchor the curve at ($t = 0$) and ($t = 1$), and two intermediate control points $P_1$ and $P_2$ that determine the curvature. The parameter $t$ interpolates smoothly between $P_0$ and $P_3$, producing a continuous and differentiable curve that always lies within the convex hull of its control points. This compact representation enables Bézier curves to capture a wide variety of nonlinear mappings with only a few parameters, making them particularly suitable for intensity transformation tasks (Magudeeswaran et al., 2021; Zhang et al., 2023). Fig. 3(a) demonstrates a set of Bézier curves and the corresponding image transformations from given source domain images (the 1st column).

As discussed above, deep learning based methods such as GAN-based style-transfer require numerous parameters and don't perform well on medical images. Meanwhile, classic methods such as histogram matching (Coltuc et al., 2006; Gonzales & Wintz, 1987) lack sufficient flexibility to be optimized for a particular source-target domain pair. Bézier curve transformation provides a simple yet powerful method for adjusting pixel intensity distributions in a controllable, non-linear manner. However, previous methods (Zhou et al., 2022; 2019) only augment data with random Bézier curve transformations. Despite the improved robustness of deep learning models, these methods cannot optimize the transformation for a specific source-target domain gap.

We propose to use Bézier curve transformation to learn the mapping between different domains. By changing the control points of the Bézier curve, we can manipulate the pixel value in the source domain and make it approximate the target domain distribution. Through the whole dataset, we find a fixed set of Bézier curve transformations that best bridge the domain gap. Two additional technical details are worth noting. First, a transformation function requires a pair of source and target images. For a source domain image, we first find its best match in the target domain, and then optimize the Bézier curve transformation so the transformed image is as close to its match as possible. Second, when comparing images, it is too limited to compare in the pixel space, especially since there is a significant distribution shift between pixel intensity distributions of the two domains. Instead, we resort to the strong representation power of deep learning models, which extract high-level structural and anatomical features that remain invariant across domains. We propose to use a pretrained encoder and compare images in the latent space. In practice, we first identify several prototype images from the source domain using the $K$-means clustering method in feature space. Subsequently, we search the target domain's feature space to identify target domain images that match the selected source domain images best. By matching the source-target image pairs, we optimize the control points by minimizing the mean squared error (MSE) loss between their extracted features.

Figure 4: An overview of the proposed uncertainty-guided CDM training framework.

Due to the structural differences between matched source and target images, the optimization function is highly noisy, making gradient-based optimization methods unstable and difficult to converge. Given that the parameters to be optimized reside in a low-dimensional space (4D), we employ the Nelder–Mead method to optimize the control points of the Bézier curves. Compared with the deep learning based style transfer method, our Bézier adaptation preserves more original image information and reduces the risk of corruption. The randomness introduced by optimizing with different matched pairs ensures enough diversity, preventing our method from being trapped in corner cases. The complete Bézier adaptation pipeline is in Fig. 3(b). Pseudo code is shown at Appendix A.

## 3.2 UNCERTAINTY-GUIDED TRAINING OF A CONDITIONAL DIFFUSION MODEL

**Diffusion models** are generative models that synthesize data by progressively denoising random noise. The forward process incrementally adds noise to training data, while the reverse process reconstructs data through denoising, described via stochastic differential equations: $dx_t = f(x_t, t)dt + g(t)dw_t$ and $dx_t = [f(x_t, t) - g(t)^2 \nabla_{x_t} \log p_t(x_t)]dt + g(t)d\bar{w}_t$, where $x_t$ is the data state at time $t$, $f(\cdot, \cdot)$ is the drift, $g(\cdot)$ the volatility, and $w_t, \bar{w}_t$ are Wiener processes. The score function $\nabla x_t \log p_t(x_t)$, essential for generation, is intractable and approximated by a score network $s_\theta$ trained via denoising score matching (Ho et al., 2020): $E_t\{\lambda(t)E_{x_0}E_{x_t|x_0}[\|s_\theta(x_t, y, t) - \nabla_{x_t} \log p_{t|0}(x_t|x_0, y)\|_2^2]\}$, with $x_0, y \sim p_0(X, Y)$ and Gaussian transition kernel $p_{t|0}(X_t|X_0, Y)$.

In this subsection, we explain how to train a target domain CDM. Recall we only have an imperfect initial segmentation model (potentially trained in the source domain). We plan to adapt such a segmentation model to target domain images and then use the pseudo-label as the conditional input to train the CDM. We propose an uncertainty-guided training method, which jointly considers all available semantic predictions from the initial segmentation model.

The basic idea is illustrated in Fig. 4. For the same input image, we generate multiple pseudo segmentation masks, each of which has a corresponding confidence map, and all of them are used for the training. These pseudo segmentation labels are Arg-Max prediction (i.e., the most possible class labels at all pixels), Arg-2nd prediction (i.e., the second most possible class labels at all pixels), Arg-3rd, Arg-4th, etc. For each of these pseudo segmentation masks, we can extract the corresponding confidence maps. The Arg-Max confidence map captures, for each pixel, the maximum confidence score across all classes. The Arg-2nd confidence map captures, for each pixel, the 2nd highest confidence score across all classes. Other confidence maps (Arg-3rd, Arg-4th, etc.) are defined accordingly. We argue that all these pseudo segmentation labels are valuable, especially when the segmentation model is not reliable enough. The rationale is that, for some uncertain region, Arg-2nd will have a big chance to be the true label if Arg-Max prediction has low confidence.

For the CDM training, we will train the score model using all these predictions as conditional input, and each prediction leads to a separate score estimation. We combine the score estimations with these different conditional masks using their corresponding pixel-wise confidence maps, resulting in our final score estimation, which is used for the score matching loss. Note that the confidence-based reweighting is pixel-wise, as at different locations of an image, the prediction confidence can be different. This uncertainty-guided score matching approach allows for more effective use of all the label information while reducing the negative impact of erroneous predictions in the pseudo-label.

Formally, given a target domain image $x \in X_t$, and an imperfect initial segmentation model $f_p$. We denote by $\tilde{y}^{(k)}$ the $k$-th prediction, and $c^{(k)}$ its corresponding $k$-th confidence map. So $\tilde{y}^1$ and $c^1$

are the Arg-Max prediction and its confidence map. $\tilde{y}^2$ and $c^2$ are the Arg-2nd prediction and its confidence map. Similarly for Arg-3rd, Arg-4th, etc. All the predictions are fed to the score network $s_\theta$ and the gradient of the conditional log likelihood, $\nabla_{x_t} \log p_t(x_t|Y)$, where $Y$ is the condition, is estimated as a weighted convex combination of the score network outputs, with weights determined by the corresponding confidence maps. We define the uncertainty reweighted score network as: $\hat{s}_\theta(x_t, \tilde{c}, t) = \sum_{k=1}^{|C|} c^{(k)} \cdot s_\theta(x_t, \tilde{y}^{(k)}, t)$. The objective function of our uncertainty guided score matching is presented as follows: $E_t\{\lambda(t)E_{x_0}E_{x_t|x_0}[\|\hat{s}_\theta(x_t, \tilde{y}^{(k)}, t) - g_t(x_0, x_t, y)\|_2^2]\}$, in which "$\odot$" is the Hadamard product (pixel-wise product), $\lambda(t)$ is the scheduling weight for time step $t$, and $g_t(x_0, x_t, y) = \nabla_{x_t} \log p_{t|0}(x_t|x_0, y)$ is the transition kernel.

Optimizing this objective function ensures that predictions with high confidence contribute sufficient semantic information to the score estimation process, while still taking into consideration the information carried by predictions with relatively low confidence/high uncertainty. This approach is particularly beneficial in regions where the prediction confidence is not biased toward a specific class.

In practice, we observe that predictions with extremely low confidence are not informative and confuse the learning procedure. To address this issue, we propose a confidence-thresholding strategy: for $k > 1$, at any pixel $(i, j)$, if the corresponding highest confidence value $c^{(1)}(i, j)$ at this location is above a predefined threshold $\delta$, we replace all the pseudo-labels at this pixel with the Arg-Max prediction, i.e, $\tilde{y}^{(k)}(i, j) = \tilde{y}^{(1)}(i, j)$, and set the corresponding confidence value $c^{(k)}(i, j) = \frac{1}{|C|}$, where $|C|$ is the number of classes. This often happens with background pixels, as well as non-background pixels, where Arg-Max has very high confidence, and we essentially trust the Arg-Max prediction completely at these pixels. The Pseudocode can be found in Appendix A.

## 4 EXPERIMENT

In the experimental section, we mainly evaluate the effectiveness of our method on unsupervised domain adaptation (UDA) for medical image segmentation.

**Datasets.** We use three benchmark datasets (i.e., BraTS 2023 dataset (Kazerooni et al., 2024), Multi-Modality Whole Heart Segmentation dataset (Zhuang & Shen, 2016), and Abdominal Multi-Organ datasets (Kavur et al., 2021; Landman et al., 2015)) to show the effectiveness of our method. Please refer to Appendix C for more details about the datasets.

**Baselines.** 'No adaptation' denotes the baseline results obtained by evaluating a segmentation model trained solely on source domain data without applying any UDA techniques. CycleGAN (Zhu et al., 2017), CyCADA (Hoffman et al., 2018), SIFA (Chen et al., 2020), and PSIGAN (Jiang et al., 2020) are representative GAN-based UDA methods. ADVENT (Vu et al., 2019) enhances UDA performance by introducing entropy minimization losses, while FDA (Yang & Soatto, 2020) leverages the Fourier Transform to reduce domain discrepancies. GenericSSL (Wang & Li, 2023) uses diffusion models to extract domain-invariant representations. FPL+ (Wu et al., 2024a) employs cross-domain data augmentation and dual-domain pseudo label generation to effectively mitigate domain shift. Diff-style Peng et al. (2023) employs a classifier-guided diffusion model for image style transfer, effectively mitigating domain gaps.

**Implementation details.** We use a U-Net with ResNet34 as the backbone for the segmentation task. The input images are augmented with random rotation, random scale, and Bézier-curve-based style augmentation. For the selected UDA method, we enhance performance by adding our generated target domain images, paired with their corresponding segmentation conditions, into the labeled training set, while also applying Bézier adaptation as an additional augmentation during training. **Ours-AD**, **Ours-GS**, and **Ours-PL** denote the integration of our proposed generated data with ADVENT (Vu et al., 2019), GenericSSL (Wang & Li, 2023), and pseudo-labeling (details in Appendix B), respectively.

### 4.1 RESULTS

We compare our *Bézier Meets Diffusion* augmented UDA methods with various baselines on three benchmark datasets and show the effectiveness of our method through the Dice coefficient (Dice) and 95% Hausdorff distance (95HD). More experimental results are shown in Appendix D.

**BraTS** is a challenging UDA dataset due to the variability of tumors across different modalities. As shown in Table 1, our method outperforms other baselines under the UDA setting. When T1 is the target domain, our *Bézier Meets Diffusion* approach significantly augments the ADVENT method,

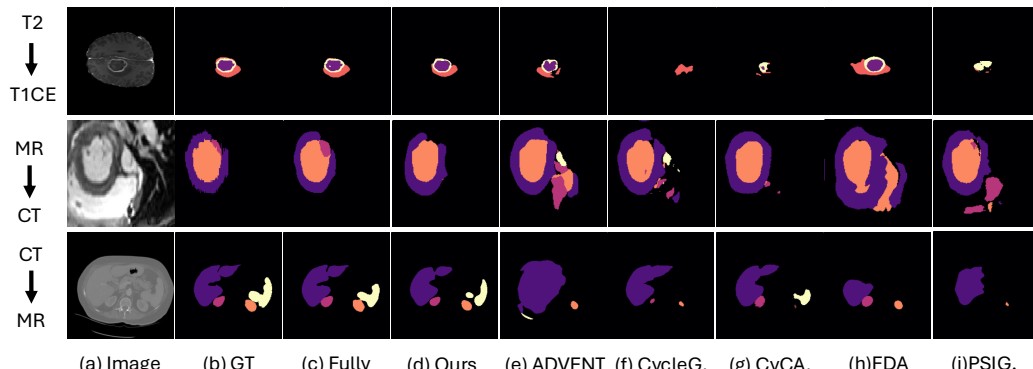

Figure 5: Segmentation results of Ours-PL and other baselines. From top to bottom: (1) BraTS dataset – Purple, orange, and yellow represent NCR, ED, and ET classes, respectively; (2) MM-WHS dataset – Purple, pink, orange, and yellow represent LVM, LAB, LVB, and AA classes, respectively; (3) Multi-Organ dataset – Purple, pink, orange, and yellow represent Liver, R. Kidney, L. Kidney, and Spleen classes, respectively.

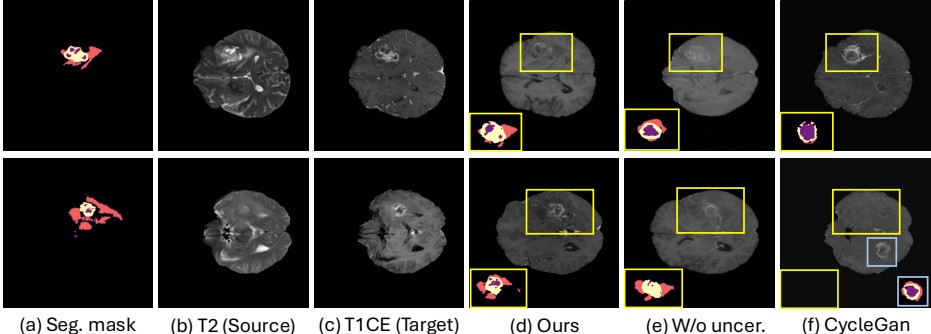

Figure 6: Qualitative results for generative models. We also show the segmentation results of the tumor region. Our method (d) is much better than baselines: CDM trained without uncertainty (e), and CycleGAN (f). In the second row, CycleGAN even hallucinates the tumor at the wrong location.

improving the Dice score by 17.0% and reducing the 95HD by 22.4%. When adapting T2 domain images to T1CE, **Ours-PL** outperforms other methods by a large margin. The generative-based baselines perform worse on this dataset because they rely on style transfer, which struggles in tumor regions due to the variability and sparsity of tumors, ultimately impacting segmentation performance.

**MM-WHS** serves as a strong benchmark for UDA methods due to the realistic domain gap between CT and MRI scans, particularly in contrast, noise characteristics, and anatomical appearance. Our method achieves impressive performance, especially in 95HD. As shown in Table 2, our *Bézier Meets Diffusion* approach boosts ADVENT (Vu et al., 2019) substantially, achieving a 19.7% improvement in Dice score and a 24.8% reduction in 95HD.

The results of **Multi-Organ** are shown in Table 3. Although ADVENT (Vu et al., 2019) achieves subpar results on this dataset, our *Bézier Meets Diffusion* pipeline significantly improves its performance, surpassing other baselines. By combining our *Bézier Meets Diffusion* with the pseudo-labeling framework, the **Ours-PL** surpasses other baselines by a large margin. GenericSSL (Wang & Li, 2023) underperforms on this dataset, while the incorporation of our synthetic target domain images significantly improves its performance, highlighting the effectiveness of our pipeline.

Overall, the performance gains achieved by our *Bézier Meets Diffusion* + UDA demonstrate the effectiveness of the proposed method. Qualitative segmentation results are presented in Fig. 5. The Qualitative synthetic results in Fig. 6 highlight the ability of our uncertainty-guided CDM to produce high-fidelity target domain images with accurate alignment to the conditional masks.

## 4.2 ABLATION STUDY

In this section, we conduct extensive ablation studies to justify the effectiveness of different components. More ablation study results about the impact of varying hyperparameters are in Appendix F.

Table 1: Comparison of different methods on the BraTS dataset.

| Target domain | T1 | | | | | | | | T1CE | | | | | | | |
|---|---|---|---|---|---|---|---|---|---|---|---|---|---|---|---|---|
| | Dice (%) ↑ | | | | 95HD ↓ | | | | Dice (%) ↑ | | | | 95HD ↓ | | | |
| Method | WT | TC | ET | Average | WT | TC | ET | Average | WT | TC | ET | Average | WT | TC | ET | Average |
| No Adaptation | 8.30 | 8.81 | 5.34 | 7.48 | 60.90 | 71.04 | 114.37 | 82.10 | 9.96 | 20.35 | 14.57 | 14.96 | 59.10 | 59.55 | 65.03 | 61.23 |
| Fully supervised | 73.30 | 61.63 | 41.63 | 58.86 | 11.74 | 23.12 | 29.49 | 21.45 | 75.70 | 83.79 | 80.37 | 79.95 | 10.85 | 8.74 | 15.23 | 11.61 |
| CycleGAN | 7.21 | 5.95 | 3.88 | 5.68 | 55.47 | 65.78 | 80.31 | 67.19 | 12.55 | 23.93 | 12.49 | 16.32 | 57.40 | 58.61 | 63.30 | 59.77 |
| CyCADA | 13.00 | 18.25 | 12.08 | 14.44 | 58.84 | 112.97 | 129.56 | 100.45 | 22.26 | 35.71 | 16.79 | 24.92 | 61.17 | 76.48 | 80.84 | 72.83 |
| ADVENT | 54.24 | 49.49 | 32.91 | 45.55 | 25.64 | 37.61 | 41.53 | 34.93 | 40.67 | 48.28 | 34.81 | 41.25 | 33.96 | 44.02 | 46.79 | 41.59 |
| FDA | 32.52 | 36.21 | 22.87 | 30.53 | 73.01 | 98.52 | 104.61 | 92.05 | 39.91 | 53.17 | 35.77 | 42.95 | 39.20 | 56.11 | 59.39 | 51.57 |
| SIFA | 55.43 | 50.72 | 33.81 | 46.66 | 23.81 | 35.68 | 39.25 | 32.91 | 42.37 | 49.74 | 35.62 | 42.58 | 31.59 | 43.24 | 45.82 | 40.22 |
| PSIGAN | 47.62 | 40.92 | 25.57 | 38.04 | 25.70 | 46.09 | 54.08 | 41.96 | 21.68 | 32.83 | 17.98 | 24.16 | 41.47 | 70.43 | 74.69 | 62.20 |
| GenericSSL | 57.99 | 53.62 | 32.93 | 48.18 | 37.50 | 36.65 | 41.28 | 38.48 | 46.07 | 55.18 | 35.30 | 45.52 | 53.41 | 46.18 | 50.91 | 50.16 |
| FPL+ | 58.97 | 54.83 | 34.68 | 49.49 | 33.41 | 34.86 | 38.05 | 35.44 | 48.72 | 56.17 | 36.63 | 47.17 | 49.06 | 45.37 | 48.22 | 47.55 |
| Diff-style | 46.11 | 46.91 | 31.25 | 41.42 | 43.41 | 46.99 | 49.08 | 46.49 | 37.69 | 45.12 | 19.18 | 34.00 | 37.60 | 44.66 | 53.62 | 45.29 |
| **Ours-AD** | 63.53 | **57.71** | **38.69** | **53.31** | **20.31** | **28.03** | **32.99** | **27.11** | 57.99 | 61.77 | 34.38 | 51.38 | **20.77** | 33.62 | 39.38 | 31.26 |
| **Ours-GS** | **65.09** | 55.55 | 38.03 | 52.89 | 41.26 | 29.53 | 35.64 | 35.48 | 46.34 | 67.35 | 44.33 | 52.67 | 57.20 | 22.39 | 25.61 | 35.07 |
| **Ours-PL** | 60.90 | 56.95 | 38.21 | 52.02 | 24.97 | 34.57 | 37.94 | 32.50 | 62.47 | 67.89 | 44.12 | 58.16 | 20.89 | 23.07 | 27.84 | 23.93 |

Table 2: Comparison of different methods on the MM-WHS dataset.

| | Cardiac MRI → Cardiac CT | | | | | | | | | | Cardiac CT → Cardiac MRI | | | | | | | | | |
|---|---|---|---|---|---|---|---|---|---|---|---|---|---|---|---|---|---|---|---|---|
| | Dice (%) ↑ | | | | | 95HD ↓ | | | | | Dice (%) ↑ | | | | | 95HD ↓ | | | | |
| Method | LVM | LAB | LVB | AA | Average | LVM | LAB | LVB | AA | Average | LVM | LAB | LVB | AA | Average | LVM | LAB | LVB | AA | Average |
| No Adaptation | 63.72 | 66.73 | 82.84 | 68.19 | 70.37 | 9.03 | 18.33 | 10.67 | 15.50 | 13.38 | 25.94 | 22.11 | 45.25 | 10.31 | 25.90 | 68.60 | 64.35 | 60.88 | 72.45 | 66.57 |
| Fully supervised | 88.33 | 89.69 | 94.70 | 83.10 | 88.95 | 2.58 | 5.01 | 8.30 | 13.30 | 7.30 | 90.94 | 94.47 | 92.98 | 96.13 | 93.63 | 3.37 | 15.37 | 3.86 | 6.26 | 7.21 |
| CycleGAN | 74.76 | 83.02 | 88.66 | 84.34 | 82.70 | 14.78 | 38.68 | 6.44 | 36.64 | 24.14 | 65.91 | 36.21 | 79.55 | 18.84 | 50.13 | 47.04 | 49.12 | 47.22 | 48.39 | 47.94 |
| CyCADA | 69.78 | 87.63 | 84.69 | 65.43 | 76.88 | 19.96 | 17.12 | 9.67 | 25.05 | 17.95 | 64.56 | 47.64 | 81.55 | 31.39 | 56.28 | 24.96 | 54.06 | 32.02 | 61.37 | 43.10 |
| ADVENT | 75.08 | 79.19 | 89.11 | 87.90 | 82.82 | 15.16 | 14.74 | 5.15 | 13.74 | 12.19 | 62.60 | 47.63 | 80.92 | 33.15 | 56.08 | 43.36 | 54.07 | 37.89 | 33.56 | 42.22 |
| FDA | 59.09 | 33.88 | 73.90 | 17.99 | 46.22 | 14.30 | 29.76 | 20.08 | 62.38 | 31.63 | 35.82 | 14.77 | 53.69 | 0.74 | 26.26 | 60.02 | 53.54 | 60.37 | 72.05 | 61.50 |
| SIFA | 65.72 | 78.36 | 81.59 | 71.82 | 74.37 | 20.18 | 27.05 | 21.73 | 26.56 | 23.88 | 51.57 | 67.15 | 78.23 | 66.07 | 65.75 | 77.08 | 55.41 | 34.35 | 77.67 | 61.13 |
| PSIGAN | 13.90 | 22.26 | 27.42 | 63.24 | 31.71 | 26.96 | 29.70 | 23.47 | 27.74 | 26.97 | 69.95 | 42.81 | 79.04 | 39.79 | 57.90 | 36.15 | 66.97 | 47.25 | 52.99 | 50.84 |
| GenericSSL | 87.95 | 91.88 | 91.02 | 91.06 | 90.48 | 5.55 | 6.05 | 4.52 | 23.29 | 9.85 | 74.16 | 55.07 | 86.13 | 54.55 | 67.48 | 37.36 | 47.11 | 39.46 | 52.22 | 44.04 |
| FPL+ | 75.50 | 81.20 | 86.40 | 78.30 | 80.35 | 12.80 | 17.50 | 9.20 | 18.30 | 14.45 | 59.84 | 54.97 | 79.12 | 44.37 | 59.58 | 28.77 | 65.42 | 38.75 | 42.34 | 43.82 |
| Diff-style | 51.71 | 36.92 | 31.83 | 64.77 | 46.31 | 17.12 | 88.67 | 22.13 | 50.13 | 44.51 | 57.30 | 25.89 | 64.69 | 3.12 | 37.75 | 53.27 | 43.19 | 55.29 | 43.96 | 48.93 |
| **Ours-AD** | 79.04 | 85.23 | 89.45 | 85.92 | 84.91 | 6.78 | 10.50 | **4.38** | **6.80** | **7.11** | 68.31 | 63.97 | **86.77** | 56.59 | 68.91 | **12.59** | 59.19 | **30.18** | 40.39 | 35.59 |
| **Ours-GS** | 86.84 | **92.04** | **92.63** | 89.07 | 90.14 | 10.20 | 6.07 | 9.27 | 19.31 | 11.21 | 63.10 | **70.56** | 86.24 | 61.66 | 70.39 | 26.96 | **30.77** | 33.57 | 47.97 | 34.82 |
| **Ours-PL** | 79.18 | 84.50 | 86.78 | 89.23 | 84.92 | 7.11 | 11.29 | 5.36 | 17.55 | 10.33 | 66.95 | 63.37 | 83.83 | 50.00 | 66.04 | 22.46 | 40.86 | 43.64 | 27.76 | 33.68 |

**Components.** We conduct experiments on the BraTS 2023 dataset to validate the effectiveness of each component of our method, with a specific focus on domain adaptation from T2 to T1CE and the mean-teacher-like pseudo-labeling framework. The results are shown in Table 4. In "w/o uncertainty", the CDM is trained by pseudo-labels obtained by applying the argmax function to the initial segmentation model output. Without our proposed training strategy, a significant performance drop is observed in target domain segmentation. "mean-teacher" denotes the result when the pseudo-labeling framework is applied without augmentation from our high-quality target image generations. It reflects the effectiveness of our strategy in boosting the pseudo-labeling method. "w/o real" shows the result of the segmentation model trained with our proposed generations without real source domain images. We observe that even without the support of real images and their ground truth, the segmentation model trained on our labeled synthetic target images achieves respectable performance in target domain segmentation. This demonstrates the high quality of our generated images.

**Preliminary adaptation.** In this paper, we employ Bézier adaptation to improve the quality of pseudo-labels and their associated confidence maps, ensuring they are sufficiently reliable to guide our strategy for training a conditional diffusion model (CDM) capable of generating high-quality target-domain images. To validate its effectiveness, we compare Bézier adaptation with alternative approaches using metrics from two perspectives: (i) the quality of confidence maps and (ii) the final UDA performance. For confidence map evaluation, we adopt metrics from the uncertainty estimation literature. The area under the risk-coverage curve (AURC) quantifies failure prediction, where a lower value indicates that correctly predicted samples receive higher confidence scores, enabling more effective filtering. Excess-AURC (E-AURC) serves as the normalized variant of AURC (Moon et al., 2020; Li et al., 2023). Calibration assesses the alignment between model confidence and the true likelihood of correctness. To this end, we use expected calibration error (ECE) (Naeini et al., 2015), the Brier score (Brier, 1950), and negative log-likelihood (NLL) to evaluate calibration quality.

As shown in Table 5, compared to 'No adaptation', Histogram Matching (Coltuc et al., 2006; Gonzales & Wintz, 1987), Fourier Domain Adaptation (FDA) (Yang & Soatto, 2020), and Random

Table 3: Comparison of different methods on the Abdominal Multi-Organ dataset.

| Method | Abdominal MRI → Abdominal CT | | | | | | | | | | Abdominal CT → Abdominal MRI | | | | | | | | | |
|---|---|---|---|---|---|---|---|---|---|---|---|---|---|---|---|---|---|---|---|---|
| | Dice (%) ↑ | | | | | 95HD ↓ | | | | | Dice (%) ↑ | | | | | 95HD ↓ | | | | |
| | Liver | R. Kid | L. Kid | Spleen | Average | Liver | R. Kid | L. Kid | Spleen | Average | Liver | R. Kid | L. Kid | Spleen | Average | Liver | R. Kid | L. Kid | Spleen | Average |
| No Adaptation | 9.66 | 70.32 | 61.45 | 17.05 | 39.62 | 97.56 | 29.58 | 37.91 | 88.32 | 63.34 | 22.12 | 57.94 | 59.52 | 56.35 | 48.98 | 72.88 | 42.06 | 40.48 | 43.23 | 49.66 |
| Fully supervised | 89.04 | 85.49 | 90.50 | 87.22 | 88.06 | 14.28 | 11.83 | 6.82 | 9.49 | 10.61 | 83.60 | 83.69 | 77.01 | 65.90 | 77.55 | 14.09 | 10.12 | 17.87 | 25.36 | 16.86 |
| CycleGAN | 74.09 | 67.88 | 62.03 | 76.24 | 70.06 | 27.66 | 28.82 | 30.97 | 20.43 | 26.97 | 57.15 | 51.62 | 55.38 | 56.59 | 55.18 | 32.69 | 34.46 | 30.31 | 26.34 | 30.95 |
| CyCADA | 63.31 | 69.60 | 72.06 | 74.09 | 69.77 | 41.27 | 23.49 | 17.83 | 20.66 | 25.81 | 53.59 | 60.18 | 60.74 | 58.70 | 58.30 | 38.03 | 35.97 | 28.08 | 31.10 | 33.29 |
| ADVENT | 21.18 | 67.98 | 72.19 | 14.95 | 44.08 | 80.76 | 31.86 | 26.98 | 90.31 | 57.48 | 20.77 | 57.94 | 59.52 | 56.35 | 48.64 | 79.51 | 42.06 | 40.48 | 43.65 | 51.42 |
| FDA | 71.55 | 81.27 | 71.02 | 80.06 | 75.97 | 29.88 | 16.56 | 26.37 | 16.61 | 22.36 | 51.16 | 64.20 | 63.90 | 67.96 | 61.81 | 47.78 | 31.80 | 32.76 | 25.40 | 34.44 |
| SIFA | 41.34 | 37.92 | 32.96 | 53.59 | 41.45 | 30.01 | 37.73 | 18.15 | 32.10 | 29.50 | 48.43 | 46.55 | 58.63 | 56.27 | 52.47 | 41.52 | 52.09 | 25.40 | 45.20 | 41.05 |
| PSIGAN | 63.53 | 76.94 | 77.09 | 71.01 | 72.14 | 34.66 | 20.41 | 19.12 | 25.91 | 25.03 | 34.20 | 51.65 | 53.99 | 58.79 | 49.66 | 60.94 | 45.34 | 36.84 | 33.58 | 44.17 |
| GenericSSL | 36.79 | 56.33 | 36.22 | 9.72 | 34.77 | 62.18 | 35.98 | 54.94 | 82.72 | 58.95 | 62.87 | 53.03 | 45.21 | 49.99 | 52.78 | 33.99 | 39.73 | 43.02 | 37.06 | 38.45 |
| FPL+ | 74.20 | 79.15 | 71.60 | 78.45 | 75.85 | 31.40 | 16.90 | 24.50 | 17.60 | 22.60 | 55.90 | 66.85 | 64.30 | 68.10 | 63.79 | 44.35 | 32.75 | 30.70 | 26.40 | 33.55 |
| Diff-style | 41.52 | 73.38 | 76.85 | 76.23 | 67.00 | 62.85 | 21.10 | 17.89 | 19.15 | 30.25 | 29.15 | 57.94 | 58.84 | 57.44 | 50.84 | 70.19 | 41.47 | 34.95 | 37.93 | 46.13 |
| **Ours-AD** | **81.03** | 84.50 | 75.30 | **88.67** | 82.38 | 20.74 | 12.45 | 22.16 | 8.82 | 16.04 | 54.56 | 67.84 | 69.80 | 75.34 | 66.89 | 35.94 | 27.95 | 26.93 | 19.84 | 27.66 |
| **Ours-GS** | 75.05 | **86.72** | 77.03 | 84.24 | 80.76 | 26.23 | **10.17** | 19.85 | 13.20 | 17.36 | **68.73** | 72.61 | **75.37** | 73.25 | **72.49** | **29.69** | 23.65 | **19.94** | 21.22 | 23.63 |
| **Ours-PL** | **84.89** | 85.74 | **82.33** | **90.47** | **85.86** | **17.31** | 10.57 | **14.35** | **7.26** | **12.37** | 59.02 | **74.83** | 70.78 | **79.44** | **71.02** | 35.25 | **19.44** | 22.17 | **14.16** | **22.76** |

Table 4: Ablation study of components.

| Target domain | T1C | | | | | | | |
|---|---|---|---|---|---|---|---|---|
| Method | Dice (%) ↑ | | | | 95HD ↓ | | | |
| | WT | TC | ET | Average | WT | TC | ET | Average |
| w/o real | 54.43 | 60.33 | 33.75 | 49.51 | 25.83 | 39.10 | 41.38 | 35.44 |
| Mean-teacher | 60.20 | 62.08 | 35.36 | 52.55 | 36.86 | 51.89 | 45.17 | 44.64 |
| w/o threshold | 57.30 | 58.13 | 30.54 | 48.66 | 32.42 | 37.81 | 44.37 | 38.20 |
| w/o uncertainty | 59.05 | 62.94 | 36.66 | 52.88 | 23.65 | 39.30 | 43.73 | 35.56 |
| **Ours-PL** | **62.47** | **67.89** | **44.12** | **58.16** | **20.89** | **23.07** | **27.84** | **23.93** |

Table 5: Ablation study of cold start methods.

| Target domain | T1C | | | | | | |
|---|---|---|---|---|---|---|---|
| Method | Accuracy (×10²) | | Uncertainty evaluation (×10³) | | | | |
| | Dice↑ | 95HD↓ | AURC↓ | E-AURC↓ | ECE↓ | NLL↓ | Brier↓ |
| No adaptation | 14.96 | 61.23 | 0.86 | 0.78 | 7.92 | 41.27 | 7.64 |
| Histogram matching | 11.98 | 64.72 | 0.78 | 0.65 | 9.61 | 62.36 | 9.78 |
| Fourier transform | 14.72 | 58.23 | 0.39 | 0.34 | 6.23 | 46.11 | 6.39 |
| Random Bézier | 41.88 | 34.87 | 0.06 | 0.05 | 2.72 | 13.65 | 3.22 |
| **Bézier adapt** | **48.45** | **30.25** | **0.05** | **0.04** | **2.34** | **11.50** | **2.79** |

Bézier (Zhou et al., 2022), our Bézier adaptation achieves superior results across both accuracy and uncertainty-based metrics.

This demonstrates that models trained with Bézier adaptation produce more reliable pseudo-labels and higher-quality confidence maps, which in turn enhance our method. We further conduct an ablation study to assess the influence of different style transfer methods on the final UDA performance. As shown in Table 6, Bézier adaptation consistently outperforms alternative approaches, underscoring its effectiveness as a style transfer strategy and confirming its role as a critical component of our framework.

**Number of confidence maps.** We conduct an ablation study on the number of confidence maps ($k$) employed for uncertainty-guided CDM training. As illustrated in Table 7, the best performance is achieved when $k = 2$. In practice, we also observe that, except for Arg-Max and Arg-2nd, other predictions contain less information (i.e., most of their confidence maps exhibit near-zero values). We may safely drop these predictions in order to save GPU memory and maximize our batch size, given limited computational resources. Also, noting that the training time for convergence increases significantly as the $k$ increases, especially when $k = 3$.

Table 6: Ablation study on the impact of different style transfer methods.

| Target domain | T1C | | | | | | | |
|---|---|---|---|---|---|---|---|---|
| Hyper-parameters | Dice (%) ↑ | | | | Hausdorff Distance (mm) ↓ | | | |
| | WT | TC | ET | Average | WT | TC | ET | Average |
| Source only | 26.88 | 25.71 | 13.88 | 22.16 | 56.21 | 98.27 | 105.10 | 86.53 |
| CycleGAN | 41.09 | 42.73 | 25.54 | 36.45 | 34.42 | 51.94 | 59.74 | 48.70 |
| CyCADA | 57.25 | 58.15 | 30.89 | 48.76 | 29.15 | 33.90 | 38.92 | 33.99 |
| PSIGAN | 34.76 | 41.97 | 19.26 | 32.00 | 31.30 | 74.24 | 75.92 | 60.49 |
| FDA | 41.81 | 45.24 | 29.12 | 38.72 | 39.01 | 61.39 | 66.79 | 55.73 |
| **Ours-Bézier** | **62.47** | **67.89** | **44.12** | **58.16** | **20.89** | **23.07** | **27.84** | **23.93** |

Table 7: Ablation study on the # of confidence maps used in uncertainty guided CDM training.

| Target domain | T1C | | | | | | | |
|---|---|---|---|---|---|---|---|---|
| Hyper-parameters | Dice (%) ↑ | | | | Hausdorff Distance (mm) ↓ | | | |
| | WT | TC | ET | Average | WT | TC | ET | Average |
| $k = 1$ | 59.05 | 62.94 | 36.66 | 52.88 | 23.65 | 39.30 | 43.73 | 35.56 |
| $k = 2$ | **62.47** | **67.89** | **44.12** | **58.16** | 20.89 | **23.07** | **27.84** | **23.93** |
| $k = 3$ | 60.57 | 65.11 | 34.01 | 53.23 | **16.83** | 29.56 | 35.96 | 27.45 |

## 5 CONCLUSION

We propose *Bézier Meets Diffusion*, a novel framework for medical image segmentation that combines Bézier-curve-based style transfer with conditional diffusion models. To mitigate domain gaps, we regularize the transfer mapping with Bézier curves and optimize their control points for more effective transformations. To handle noisy pseudo-labels during CDM training, we introduce an uncertainty-guided score matching strategy that leverages multiple uncertainty-weighted predictions from the segmentation model. By combining these two strategies, our method effectively augments training data and achieves improved segmentation performance across multiple UDA benchmarks. Furthermore, *Bézier Meets Diffusion* can be seamlessly integrated into existing UDA pipelines to further enhance their performance.

# 6 ETHICS AND REPRODUCIBILITY STATEMENT

## 6.1 ETHICS STATEMENT

This work focuses on developing a novel diffusion-augmented framework for medical image segmentation. All experiments are conducted on publicly available, de-identified datasets. No additional human or animal data were collected, and therefore no additional IRB approval was required.

## 6.2 REPRODUCIBILITY STATEMENT

Details of the datasets are provided in Section 4 (**Datasets**) and Appendix C, and the baselines are described in Section 4 (**Baselines**). Implementation details are presented in Section 4 (**Implementation details**) and Appendix G. We additionally include pseudo-code for uncertainty-guided score matching and Bézier adaptation in Appendix A, hyper-parameter settings with ablation studies in Appendix F, and the details of pseudo-labeling implementation in Appendix B.

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

ACKNOWLEDGMENTS

We use LLM to improve the writing quality and grammar check of the manuscript.

OVERVIEW

We include extended experimental results, implementation details, pseudo code, errata, limitations, and broader impact to support the main findings.

## A  PSEUDO CODE

In this part, we provide the pseudo code for uncertainty-guided score matching in Algorithm 1 and Bézier adaptation in Algorithm 2.

---

**Algorithm 1:** Training with uncertainty guided score matching

**Input:** Target image set $X_t$, pseudo-labeling network $f_p$, transition kernel $p_{t|0}$, confidence threshold $\delta$
**Output:** Conditional score network $s_\theta$
1 **while** *not converged* **do**
2      Sample $x_0 (N \times N)$ from $X_t$, and $t$ from $[0, T]$;
3      Sample $x_t$ from $p_{t|0}$;
4      Compute pseudo-labels and uncertainties:
       $c^{(k)}, \tilde{y}^{(k)} \leftarrow f_p(x_0), k \in C$;
5      **for** $(i, j) \in \{0, \dots, N\} \times \{0, \dots, N\}$ **do**
6          **if** $c_{ij}^{(1)} > \delta$ **then**
7             $\tilde{y}_{ij}^{(k)} = \tilde{y}_{ij}^{(1)}, c_{ij}^{(k)} = 1/|C|, \forall k \in C$
8          **end**
9      **end**
10      Update $\theta$ by a gradient descent step on
       $\nabla_\theta \| \sum_{k=1}^{|C|} c^{(k)} s_\theta - g_t(x_0, x_t, y)\|_2^2$
11 **end**

---

**Algorithm 2:** Bézier adaption

**Input:** Source image set $X_s$, Target image set $X_t$, Pretrained image feature extractor $\phi$,
**Output:** Optimized control point sets $P_1^*, P_2^*$
1 Extract Source image features $X_s^f = \phi(X_s)$ ;
2 Extract Target image features $X_t^f = \phi(X_t)$ ;
3 Apply K-means clustering on $X_s^f$ with $n_s$ clusters to find cluster centers $\{c_1, c_2, \dots, c_{n_s}\}$;
4 **for** $j \leftarrow 1$ **to** $n_s$ **do**
5      Select the closest image (feature) $x_j \in X_s$ to the cluster center $c_j$ as the prototype;
6      Find the closest image (feature) $x_p \in X_t$ to the $x_j$ as the matched target image ;
7      **Initial** random control points and transfer source domain image with Bézier curve as $B(x_j)$ ;
8      Define **objective function** as $f_l = \|\phi(B(x_j)) - \phi(x_p)\|_2^2$ ;
9      **Optimize** control points $P_1, P_2$ with Nelder–Mead method
10 **end**

---

## B  PSEUDO-LABELING DETAILS

The pseudo-labeling framework used in this paper is mean-teacher-like. We have two models: the student model and the teacher model. The parameters of the teacher model are updated by the EMA method from the student model. The teacher model is for generating pseudo-labels and confidence maps for the training of the student model on the target domain. Considering the imperfection of pseudo-labels, we filter out the low-quality pseudo-labels by thresholding on confidence maps. Only the pseudo-labels with high enough confidence are kept and used for supervising the student model. During training, as the quality of pseudo-labels progressively improves, we gradually relax the threshold to incorporate more high-quality pseudo-labels into the learning process. This threshold decreases from $\tau_u$ to $\tau_l$ during training. We supervise the training of the student model using both real labels and selected pseudo-labels, with a combination of cross-entropy and Dice losses. The loss contribution from pseudo-labels is scaled by a weighting factor $\lambda$.

## C  DATASET DETAILS

**The Brain Tumor Segmentation (BraTS) datasets** focus on brain tumor sub-region segmentation. The sub-regions are composed of enhancing tumor (ET), the tumor core (TC), and the whole tumor (WT). There are multiple modalities available in the Brats dataset, i.e., T2, Flair, T1, and T1CE. In this paper, we consider the domain adaptation from T2 to the other 3 modalities. The regions of the GD-enhancing tumor (ET — label 3), the peritumoral edematous/invaded tissue (ED — label 2), and the necrotic tumor core (NCR — label 1) are annotated in images. The TC class comprises regions ET and NCR. The WT class comprises regions ET, ED, and NCR. The training set is constructed by 142600 images (71300 source domain images, 71300 target domain images), and the validation set

Table 8: Comparison of different methods on the BraTS dataset.

| Target domain | Flair | | | | | | | |
|---|---|---|---|---|---|---|---|---|
| Method | Dice (%) ↑ | | | | 95HD ↓ | | | |
| | WT | TC | ET | Average | WT | TC | ET | Average |
| No Adaptation | 76.29 | 60.04 | 34.40 | 56.91 | 16.69 | 25.25 | 35.40 | 25.78 |
| Fully supervised | 87.33 | 68.18 | 47.81 | 67.78 | 6.12 | 17.78 | 24.19 | 16.03 |
| CycleGAN | 74.77 | 59.11 | 32.14 | 55.34 | 22.85 | 29.64 | 33.66 | 28.71 |
| CyCADA | **80.75** | 61.30 | 29.78 | 57.28 | **12.02** | 23.75 | 37.57 | 24.44 |
| ADVENT | 75.95 | 60.16 | 37.52 | 57.88 | 14.38 | 22.11 | 26.44 | 20.98 |
| FDA | 75.46 | 62.85 | 38.44 | 58.92 | 20.67 | 30.86 | 41.47 | 31.00 |
| SIFA | 69.18 | 58.11 | 39.51 | 55.60 | 19.66 | 38.79 | 49.24 | 35.90 |
| PSIGAN | 74.26 | 57.82 | 36.60 | 56.23 | 16.54 | 26.81 | 34.13 | 25.82 |
| GenericSSL | 41.73 | 52.65 | 28.72 | 41.03 | 56.88 | 34.47 | 40.26 | 43.87 |
| FPL+ | 74.66 | 54.31 | 36.73 | 55.23 | 20.40 | 30.19 | 41.04 | 30.54 |
| **Ours-AD** | 73.74 | 62.30 | 39.80 | 58.61 | 12.44 | **18.58** | **25.42** | **18.81** |
| **Ours-GS** | 73.86 | 59.66 | 36.08 | 56.53 | 24.43 | 27.45 | 32.61 | 28.17 |
| **Ours-PL** | 71.70 | **65.42** | **41.00** | **59.37** | 22.59 | 19.98 | 25.89 | 22.82 |

is constructed by 12400 source domain images. The test set is constructed by 38905 target domain images. We generate 30514 target domain synthetic images using segmentation masks from the source domain.

**The Multi-Modality Whole Heart Segmentation (MM-WHS) datasets** consist of 20 unpaired volumetric cardiac images, including CT and MRI scans. The annotations for the segmentation of four cardiac structures are provided: the left ventricle myocardium (LVM), left atrium blood cavity (LAB), left ventricle blood cavity (LVB), and ascending aorta (AA). The training set is constructed by 2304 source domain and 2016 target domain images. The validation set is constructed by 576 source domain images. The test set is constructed by 867 target domain images. For this dataset, we generate 2304 target domain synthetic images conditioning on masks from the source domain.

**The Abdominal Multi-Organ datasets** contain two modalities: MRI images from the ISBI 2019 CHAOS Challenge dataset (Kavur et al., 2021) and CT images from the Multi-Atlas Labeling Beyond the Cranial Vault - Workshop and Challenge (Landman et al., 2015). Four abdominal organs are labeled for segmentation: liver, right kidney (R. Kid), left kidney (L. Kid), and spleen. For MRI to CT, the training set comprises 491 source domain and 2449 target domain images. The validation set comprises 30 source domain images. The test set comprises 1048 target domain images. For CT to MRI, the training set comprises 2449 source domain and 491 target domain images. The validation set comprises 282 source domain images. The test set comprises 126 target domain images. For CT-to-MRI adaptation, we generate 2,449 synthetic MRI images conditioned on source domain segmentation masks. Similarly, for MRI-to-CT adaptation, we generate 1,964 synthetic CT images using the same conditioning strategy.

# D  EXTRA RESULTS

## D.1  EXTRA RESULTS ON BRATS FROM T2 TO FLAIR

We provide additional results on the BraTS dataset for the domain adaptation task from T2 to FLAIR. As shown in Table 8, our proposed method outperforms all baseline approaches in both Dice score and 95HD, demonstrating its superior segmentation performance.

## D.2  EXTRA QUALITATIVE RESULTS

Here we provide some extra qualitative results in Fig. 7. Our method achieves better performance than other baselines.

Table 9: Ablation study on the U-Net architecture.

| Target domain | T1C | | | | | | | |
|---|---|---|---|---|---|---|---|---|
| Hyper-parameters | Dice (%) ↑ | | | | Hausdorff Distance (mm) ↓ | | | |
| | WT | TC | ET | Average | WT | TC | ET | Average |
| ours-ResNet18 | 60.12 | 63.32 | 36.93 | 53.46 | 24.48 | 40.51 | 44.17 | 36.38 |
| ours-ResNet34 | 62.47 | **67.89** | **44.12** | **58.16** | 20.89 | **23.07** | **27.84** | **23.93** |
| ours-ResNet50 | **62.56** | 66.11 | 42.83 | 57.17 | **20.32** | 25.54 | 31.78 | 25.88 |

# E   ABLATION STUDY

We present experiments to examine the impact of backbone architecture on final performance. As shown in Table 9, our method remains robust across different U-Net backbones.

# F   HYPERPARAMETERS

From Algorithm 1, we know that the confidence threshold $\delta$ is critical for the conditional generation of high-quality target images. Here, we study the effect of $\delta$ in augmenting domain adaptation training. Our empirical results (Table 10) show that our method is robust to different values of $\delta$. As described in Appendix B, hyperparameter $\lambda$ is the loss weight of pseudo-labeling supervision. The hyperparameters $\tau_l$ and $\tau_u$ are the lower and upper bounds of the confidence threshold for filtering out low-quality pseudo-labels during training. Due to the high-quality generations in the target domain, the performance of the pseudo-labeling framework is robust to the perturbation of $\lambda$, $\tau_l$, and $\tau_u$. This reflects the ability of our method in conditionally generating high-quality target domain images with pseudo conditions.

Table 10: Ablation study results of hyperparameters on the BraTS dataset. Overstriking hyperparameter values is our setting.

| Target domain | T1C | | | | | | | |
|---|---|---|---|---|---|---|---|---|
| Hyper-parameters | Dice (%) ↑ | | | | Hausdorff Distance (mm) ↓ | | | |
| | WT | TC | ET | Average | WT | TC | ET | Average |
| $\delta$-0.7 | **62.52** | 66.30 | 42.06 | 56.96 | 25.25 | 36.54 | 41.54 | 34.45 |
| $\delta$-**0.8** | 62.47 | **67.89** | **44.12** | **58.16** | 20.89 | **23.07** | **27.84** | **23.93** |
| $\delta$-0.9 | 60.15 | 63.41 | 38.75 | 54.10 | 30.06 | 38.54 | 44.71 | 37.77 |
| $\lambda$-0.6 | 62.69 | 66.57 | 42.86 | 57.37 | 25.19 | 30.95 | 35.87 | 30.67 |
| $\lambda$-**0.7** | 62.47 | **67.89** | **44.12** | **58.16** | 20.89 | **23.07** | **27.84** | **23.93** |
| $\lambda$-0.8 | **63.96** | 67.45 | 39.69 | 57.03 | 24.57 | 29.14 | 34.28 | 29.33 |
| $\tau_l$-0.4 | **62.94** | **68.11** | 39.01 | 56.69 | 25.24 | 26.31 | 31.61 | 27.72 |
| $\tau_l$-**0.5** | 62.47 | 67.89 | **44.12** | **58.16** | 20.89 | **23.07** | **27.84** | **23.93** |
| $\tau_l$-0.6 | 62.13 | 67.16 | 40.68 | 56.65 | 21.05 | 24.12 | 30.36 | 25.18 |
| $\tau_u$-0.4 | 61.24 | 65.01 | 39.03 | 55.09 | 24.92 | 32.31 | 37.20 | 31.48 |
| $\tau_u$-**0.7** | 62.47 | **67.89** | **44.12** | **58.16** | 20.89 | **23.07** | **27.84** | **23.93** |
| $\tau_u$-0.8 | **62.90** | 67.94 | 40.39 | 57.07 | **18.44** | 25.97 | 32.16 | 25.52 |

# G   IMPLEMENTATION DETAILS

We train the segmentation model for 200 epochs with a batch size of 64. The segmentation network is optimized by Adam with a learning rate of 1e-4. The diffusion model is trained for 250,000 iterations with a batch size of 22, using two NVIDIA H100 80GB HBM3 GPUs, 46 Intel(R) Xeon(R) Platinum 8462Y+ CPUs, and 2,062,607 MB of RAM.

The pretrained feature extractor used in Algorithm 2 is adopted from the autoencoder in Stable-Diffusion-v1-4 [1].

---

[1] https://huggingface.co/CompVis/stable-diffusion-v1-4

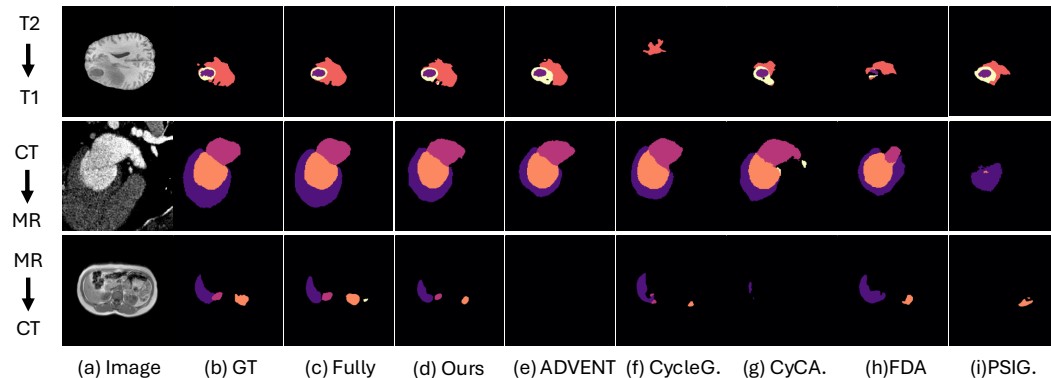

Figure 7: Segmentation results of ours (Ours-PL) and other baselines.

In Tab. 4, "w/o threshold" is the ablation result without using the thresholding strategy described in the main text from line 195 to line 205, which uses Arg-Max in high confidence regions as the pseudo-labels for CDM training. This result reflects the effectiveness of this well-designed strategy.

## H LIMITATIONS

One limitation of our proposed method is its increased demand for GPU memory and computational resources during diffusion model training compared to training without the uncertainty guidance component.

## I BROADER IMPACT

This work proposes *Bézier Meets Diffusion*, a diffusion-based framework for unsupervised domain adaptation (UDA) in medical image segmentation. The potential positive societal impacts of this work include expanding access to high-quality medical AI tools in resource-limited healthcare settings by reducing the need for costly expert annotations. By enabling models to adapt more effectively across institutions and scanners, the method could help promote equity in diagnostic accuracy and support broader deployment of AI-assisted diagnostic tools.

