# OpenReview forum: "Bézier Meets Diffusion: Robust Generation Across Domains for Medical Image Segmentation"
_ICLR.cc/2026/Conference — Submitted to ICLR 2026_

### Official Review · Reviewer_avBb · 2025-10-26

**Soundness:** 3
**Presentation:** 3
**Contribution:** 3
**Rating:** 4
**Confidence:** 3

**Summary:**

The paper proposes “Bézier Meets Diffusion,” a UDA framework for medical image segmentation that combines a Bézier-curve-based style transfer with an uncertainty-guided conditional diffusion model (CDM).

Bézier adaptation learns low-parameter, nonlinear intensity mappings between modalities by optimizing cubic Bézier control points.

A CDM is trained on target-domain images conditioned on pseudo-labels from an initial model, with a new uncertainty-guided score matching loss that fuses Arg-Max and secondary predictions via pixel-wise confidence maps.

Extensive experiments on BraTS, MM-WHS, and abdominal datasets show improved Dice and 95HD over GAN-based and recent UDA baselines, with ablations validating each component.

**Strengths:**

Originality: The combination of feature-optimized Bézier intensity transfer with uncertainty-weighted CDM training is novel and well-motivated, bridging deterministic style mapping and probabilistic generation. The uncertainty-guided score fusion across top-k predictions is a simple yet clever way to exploit predictive distributions without architectural changes.

Quality: The experimental evaluation is thorough across three benchmarks with diverse baselines, clear ablations (components, k, thresholds, backbones), and uncertainty metrics (AURC, ECE, NLL) supporting the cold-start benefits. Methodological details (pseudo-code, hyperparameters, datasets) enable reproducibility and clarify design choices.

Clarity: The paper articulates the limitations of GAN transfer for sparse/heterogeneous lesions and motivates Bézier constraints and uncertainty fusion clearly with helpful figures. The training/inference distinction for CDM and the role of confidence thresholding are explained precisely.

Significance: Showing large gains in difficult cross-modality medical UDA (e.g., T2->T1CE) is impactful, and the approach is modular to integrate with existing UDA pipelines. The ability to synthesize labeled target-domain data at scale addresses a central bottleneck in medical imaging.

**Weaknesses:**

Dependence on feature encoder: The feature-space optimization relies on a pretrained Stable Diffusion autoencoder; potential bias/mismatch to medical features is not deeply analyzed. An ablation comparing different medical encoders or self-supervised encoders, and sensitivity to encoder choice, would improve robustness claims.

Computational cost and practicality: The uncertainty-guided CDM increases memory/time (acknowledged), but the paper lacks wall-clock comparisons against strong non-generative UDA methods under equal budgets. Providing training/inference cost-benefit analyses and lighter variants (e.g., k=2 default with memory-saving tricks, 2D vs. 3D trade-offs) would aid practical adoption.

Generalization scope: While results cover three datasets, it remains unclear how the Bézier mapping handles more complex modality shifts in 3D volumetric situation. Including evaluations on 3D medical UDA benchmark would strengthen claims.

**Questions:**

How sensitive is Bézier adaptation to the choice/number of source prototypes and target matches, and could you provide guidance or an adaptive selection strategy?

Have you evaluated different feature encoders (e.g., nnU-Net encoder, medical SSL models) for the Bézier optimization, and how does encoder choice affect both transfer quality and downstream segmentation?

Can the uncertainty-guided CDM extend to 3D volumes with spatially consistent conditioning and does k=2 remain optimal in 3D given memory constraints?

---

### Official Review · Reviewer_jYQr · 2025-10-28

**Soundness:** 2
**Presentation:** 3
**Contribution:** 1
**Rating:** 2
**Confidence:** 5

**Summary:**

This paper proposes an unsupervised domain adaptation framework that combines Bézier curve style transfer and conditional diffusion models to address the issue of cross-domain robustness in medical image segmentation.
Its main contributions are:
1) Propose a Bézier curve style transfer method guided by feature space similarity, effectively reducing the style discrepancies between different imaging modalities.
2) Build a new UDA segmentation framework and use CDM to generate annotated synthetic images to enhance the training on the target domain.
3) Develop uncertainty-guided learning strategies to enhance the training of CDM.
4) The presented framework can be combined with existing methods to improve performance.

**Strengths:**

This paper innovatively combines Bézier curve style transfer with an uncertainty-guided conditional diffusion model, significantly improving the generation quality and domain adaptation performance in cross-modal medical image segmentation.

**Weaknesses:**

1) Inconsistent Claims: The authors claim in Section 2 that their framework can generate "our framework leverages the generative capacity of CDM, enabling us to generate an unlimited number of labeled target domain images, irrespective of the availability of target domain samples." However, as shown in Appendix C, the method still relies heavily on a large amount of target domain data, and the quantity of generated data is constrained by that of source domain data.
2) Unclear Methodology: The training strategy and learnable parameters for the Bézier-curve-based style transfer module remain unclear. Furthermore, this paper does not clearly describe how the CDM enhances the segmentation model's training.
3) Weak Contributions: In Table 4, the "Mean-teacher" baseline outperforms other methods shown in Table 1, including "Ours-AD" and "Ours-GS". This highlights the effectiveness of the employed pseudo-label framework but weakens the contributions of the proposed components.
4) Unfair Comparison: The comparison in Table 6 is unfair since the competing methods are not integrated with the pseudo-label framework.
5) Lack of Metrics: There are no quantitative metrics (e.g., PSNR, SSIM, and FID) to evaluate the quality of the generated images, making it impossible to assess the predictions of the proposed generative framework.
6) Incomplete Comparison with SOTA: The experiment results lack a comparison with state-of-the-art UDA methods.

**Questions:**

1) Please revise the claim in Section 2 for consistency.
2) For clarity, please elaborate on the methodology.
3) In Table 6, please display the results of other methods combined with the pseudo-label framework.
4) Please select 1-2 quantitative metrics to evaluate the generation performance of the proposed method compared to other generative methods.

---

### Official Review · Reviewer_YKJW · 2025-10-29

**Soundness:** 2
**Presentation:** 3
**Contribution:** 2
**Rating:** 4
**Confidence:** 4

**Summary:**

The paper proposes a cross-domain generative framework for medical image segmentation, which integrates Bézier curve–based style transfer with a conditional diffusion model. The overall pipeline includes three main components: (1) employing a nonlinear Bézier transformation to perform style transfer between different domains and reduce modality gaps; (2) applying confidence-based weighting to mitigate noise in pseudo labels; and (3) using the generated target-style samples to enhance the segmentation training.

**Strengths:**

The proposed framework achieves consistently strong and impressive improvements across multiple benchmarks, showing clear effectiveness.

**Weaknesses:**

The manuscript falls short on theoretical analysis, relying on performance improvements rather than mechanistic justification. This limitation reduces the amount of insight offered to readers and may make it less suitable for publication at ICLR.

Lack of Novelty:
1. Bézier parameterization itself is not a new concept; several prior works have explored mapping-based domain adaptation.  It would benefit from including relevant studies like Su, Zixian, et al. "Rethinking Data Augmentation for Single-Source Domain Generalization in Medical Image Segmentation," presented at the AAAI Conference on Artificial Intelligence in 2023.

2. The novelty claimed largely rests on using Bézier curves for continuous and smooth mapping paths. However, there is a lack of detailed theoretical analysis or empirical validation to substantiate the effectiveness of such a differentiable parameterization. More comprehensive evidence or reasoning is needed to justify its proposed benefits.

3.The paper lacks theoretical grounding for the "uncertainty-guided conditional diffusion model" proposed. It does not thoroughly explore whether the model adheres to the theoretical framework of conditional diffusion models, addressing key aspects such as convergence and variational lower bounds. For instance, it remains unclear if using the modified objective function ensures convergence to a meaningful data distribution or represents a well-defined, reversible stochastic differential equation.

Concerns on the Experimental Section:
1. The comparison studies do not include advancements from 2024 to 2025, failing to discuss or experiment with more recent methods. Including such comparisons would provide a more current context for the paper’s contributions.

2.  As the proposed framework involves generative modeling, it lacks quantitative evaluation of the generated images, such as FID or LPIPS scores. Without these metrics, it is difficult to assess the visual quality and actual contribution of the generation module.

3. Further clarification on additional hyperparameters and computational steps is needed.  Although the uncertainty-guided strategy is sophisticated, it introduces additional hyperparameters (e.g., confidence threshold \delta in Algorithm 1) and computational steps like processing multiple predictions and confidence maps. The robustness of the approach might be overly dependent on careful tuning of these hyperparameters.

**Questions:**

Authors are suggested to clarify the effectiveness of the Bézier parameterization and add the recent baselines (See weaknesses).

---

### Official Review · Reviewer_ErFD · 2025-11-01

**Soundness:** 2
**Presentation:** 3
**Contribution:** 1
**Rating:** 0
**Confidence:** 5

**Summary:**

This paper focuses on the unsupervised domain adaptation (UDA) of medical image segmentation. The UDA problem in the context of medical image segmentation has been very widely explored. The main components of this paper include (1) a Bézier-curve-based style transfer method, guided by feature-space similarity, to reduce domain gaps across different imaging modalities (2)  a novel UDA segmentation framework that leverages a CDM to generate labeled synthetic images. (3) uncertainty-guided label correlation.

**Strengths:**

- The UDA problem in the context of medical image segmentation has been very widely explored, therefore new insights need be proposed.

- The presentation of this paper is good.

- The experimental results look promising.

**Weaknesses:**

**Overall, I think the novelty of this paper is quite incremental, where several widely-adopted techniques, such as Bézier-curve-based style transformation, conditional diffusion model and uncertainty-based label correction, have been used by previous works.**

- I think the title needs to be modified, where this term of "Robust Generation Across Domains" looks like a domain generalization task. I think the title can highlight this paper narrows to the UDA task.

- Bézier-curve-based style transformation: this technique has been used by the paper namely "Generalizable Cross-modality Medical Image Segmentation via Style Augmentation and Dual Normalization, CVPR, 2022".

- Conditional diffusion model of UDA-based medical image segmentation: please see this paper Y. Zhao, K. Lu, S. Wang, J. Lu and X. Jian, "Unsupervised Domain Adaptation for Medical Image Segmentation with Conditional Diffusion Model," 2024 IEEE International Symposium on Biomedical Imaging (ISBI), Athens, Greece, 2024, pp. 1-5, doi: 10.1109/ISBI56570.2024.10635862.

- Uncertainty-based label correction: please see this paper UPL-SFDA: Uncertainty-aware Pseudo Label Guided Source-Free Domain Adaptation for Medical Image Segmentation, TMI, 2023.

- This paper combines these techniques and lacks obvious novelties to the community. It is very hard to acknowledge the contribution of this paper in a current version.

**Questions:**

Please see the weakness.

---

### Meta-Review · Area_Chair_KoUd · 2025-12-07

**Summary:**

This work studied the UDA of medical image segmentation, using a generation-based method. Four reviewers thoroughly reviewed this paper, with 1 strong reject (Rating 0), 1 reject (Rating 2), and 2 marginally below the acceptance threshold (Rating 4). The reviewers raised some reasonable concerns, including:

Reviewer ErFD

1.	This work is incremental. I have carefully checked the references that the reviewer suggested. These works are indeed related to this paper, demonstrating that the idea of this work may not be new.

Reviewer YKJW

1.	Lack of Novelty.

2.	Lacks theoretical grounding.

3. Concerns of experiments.

Reviewer jYQr

1.	Inconsistent Claims.

2.	Unclear Methodology.

3.	Weak Contributions.

4. Unfair and incomplete comparison.

Reviewer avBb

1.	Analysis of feature encoder.

2.	Computational cost and practicality.

3.	Generalization scope.

**During the rebuttal period, the authors did not respond to the reviewers’ comments.** Based on the unsolved concerns and unclear contributions, I recommend rejecting this paper.

**Reviewer Concerns:**

All the concerns are not addressed.

**Reviewer Scores:**

The authors did not respond to the reviewers. Thus, I guess no reviewers would change the ratings.

---

### Decision · Program_Chairs · 2026-01-26

Reject